# MRI- and CT-determined changes of dysphagia / aspiration-related structures (DARS) during and after radiotherapy

Steffi U. Pigorsch[1]*, Chaline May[1], Kerstin A. Kessel[1,2], Simone Graf[3], Henning Bier[3†], Fridtjof Nüsslin[1], Birgit Waschulzik[4], Stephanie E. Combs[1,2]

1 Department of Radiation Oncology, Technical University of Munich (TUM), Munich, Germany, 2 Institute of Radiation Medicine (IRM), Helmholtz Zentrum München, Neuherberg, Germany, 3 Department of Otorhinolaryngology, Technical University of Munich (TUM), Munich, Germany, 4 Institute of Medical Informatics, Statistics and Epidemiology, Technical University of Munich (TUM), Munich, Germany

† Deceased.
* steffi.pigorsch@tum.de

**Data Availability Statement:** All relevant data are within the manuscript and its Supporting Information files.

## Abstract

### Purpose

The concept of dysphagia/aspiration-related structures (DARS) was developed against the background of severe late side effects of radiotherapy (RT) for head and neck cancer (HNC). DARS can be delineated on CT scans, but with a better morphological discrimination on magnetic resonance imaging (MRI). Swallowing function was analyzed by use of patient charts and prospective investigations and questionnaires.

### Method

Seventeen HNC patients treated with intensity-modulated radiotherapy (IMRT) ± chemotherapy between 5/2012 – 8/2015 were included. Planning CT (computed tomography) scans and MRIs (magnetic resonance imaging) prior, during 40 Gray (Gy) radiotherapy and posttreatment were available and co-registered to delineate DARS. The RT dose of each DARS was calculated. Five patients were investigated posttreatment for swallowing function and assessed by means of various questionnaires for quality of life (QoL), swallowing, and voice function.

### Results

By retrospective comparison of DARS volume, a significant change in four of eight DARS was detected over time. Three increased and one diminished. The risk of posttreatment dysphagia rose by every 1Gy above the mean dose (D mean) of RT to DARS. 7.5 was the risk factor for dysphagia in the first 6 months, reducing to 4.7 for months 6-12 posttreatment. For all five patients of the prospective part of swallowing investigations, a function disturbance was detected. These results were in contrast to the self-assessment of patients by questionnaires. There was neither a dose dependency of D mean DARS volume changes over time nor of dysphonia and no correlation between volume changes, dysphagia or dysphonia.

**Funding:** The author(s) received no specific funding for this work.

**Competing interests:** The authors have declared that no competing interests exist.

## Conclusion

Delineation of DARS on MRI co-registered to planning CT gave the opportunity to differentiate morphology better than by CT alone. Due to the small number of patients with complete MRI scans over time, we failed to detect a dose dependency of DARS and swallowing and voice disorder posttreatment.

## Introduction

Worldwide, 686.000 people were newly diagnosed with head and neck cancer (HNC) in 2012. Approximately 376.000 people died of HNC the same year [1]. Standard of care of small tumors consists of surgery or radiotherapy (RT). In locally advanced cases, surgery is complemented by radiotherapy or concomitant chemo-irradiation. If there is no possibility of tumor resection or the patient refuses surgery, chemo-irradiation is the treatment of choice. Intensity Modulated RadioTherapy (IMRT) is standard of care for radiooncological HNC treatment. Imaging integration into linear accelerators, so-called image guided radiotherapy (IGRT), and continuously improved imaging techniques before and during treatment has extended the therapeutic window. With modern techniques, the reduction of radiation-induced side effects is the second main goal besides tumor control. Radiation oncologists distinguish two types of radiation-induced side effects: acute onset, day 1 –day 90, of radiotherapy and late or chronic onset from day 91 for the remainder of life [2]. Late side effects may detrimentally influence quality of life (QoL) of HNC patients.

Irregular shaped HNC volumes and the surrounding normal tissue also called organs at risk (OAR) benefit from the development of IMRT and IGRT technology [3–5]. Due to the steep dose gradients between the Planning Target Volume (PTV) and OAR, differences between planned and delivered dosage may have different consequences than in conventional radiotherapy (RT) [6]. By IGRT, it is possible to reduce safety margins for set-up, which leads to lower dose exposure of OARs. Adaptive planning compensates anatomical changes and is realized during the course of RT (i.e. shrinkage of tumor, nodal masses, weight loss and resolving postoperative changes/edema) [7, 8]. When OAR doses can be minimized, this should be reflected in patients´ reported QoL and better OAR-function after IMRT/IGRT, as shown for salivary glands early in the IMRT era [9, 10].

The swallowing organs are situated within all of the mentioned structures (tumor, organs at risk) both inside and outside, but near the target volume for RT planning.

Eisbruch et al. developed the concept of dysphagia aspiration related structures (DARS) [11]. DARS are comprised of base of tongue, floor of mouth, superior and middle pharyngeal constrictor as well as crico-pharyngeal muscles, and the proximal part of the esophagus. The supraglottic and transglottic larynx are voice structures and build the barrier for aspiration and penetration to the lungs. In 2002, Eisbruch et al. published data on objective assessment of dysphagia with a detailed explanation of pathophysiology of deglutition disorders after concurrent chemo-irradiation [12].

IGRT is usually carried out using CT imaging. The novel approach of magnetic resonance guided radiotherapy (MRgRT) involving integration of static magnetic resonance imaging (MRI) into linear accelerators is a topic of current research. However, with interfractional repetitive static MRI during the course of treatment, it might be possible to explore the principle advantages of MRgRT. The aim of this work was to assess the changes of DARS during and after radiooncological treatment of HNC by co-registration of static MRI and planning CT scans for better delineation of DARS in order to detect morphological changes over the course

of RT. The second aim was to compare these dosimetric and volumetric data to results of functional investigation of swallowing and voice.

## Material and methods

At the department of radiooncology, University Hospital of the Technical University of Munich (TUM), 780 patients with head and neck cancer received various IMRT treatments between 1 January 2008 and 31 December 2015. Patients in this analysis were scheduled for a follow-up by 31 December2016. Thirteen patients were alive at the time of retrospective data acquisition and analysis. For prospective investigation, all surviving patients were invited. Only five patients consented to swallowing and speech tests. Eight patients refused participation.

Every patient with head and neck cancer treated at our department is examined prior to RT planning by means of MRI. The MR images are co-registered to the contrast enhanced planning CT-scan (3mm slice thickness). Patients are fixed for head and neck cancer RT with a 5-point thermoplastic mask system, which is individually adjusted to the patient. For this retrospective analysis, patients with MRI scans from three different points of time were included, representing all DARS recommended by Christianen et al. [13]. Patients with every kind of head and neck cancer, adjuvant or definitive treatment intention, independent of age, sex, and race were enrolled. Only 17 patients fulfilled this inclusion criteria. Upon commencement, there were no dose constraints concerning DARS implemented into the RT plan optimization for head and neck cancer. Thirteen of the seventeen patients were treated by definitive, while four of the seventeen received adjuvant RT with concomitant chemotherapy between 1 May 2012 and 31 August 2015. All had static MRIs performed prior to RT (MRI 1), at 40Gy (MRI 2) and six weeks post RT (MRI 3). For patient characteristics, see Table 1.

Five patients were treated by a simultaneously integrated boost (SIB) concept (adjuvant n = 1 and definitive n = 4, 1/4 after R2 excision of lymph node metastasis for cancer of unknown primary (CUP) syndrome. One patient was re-irradiated due to local recurrence prior to swallowing investigations.

The static MRI scans were performed for MRI 1 (on average 16 days prior to RT), MRI 2 (during RT at 40 Gy) and MRI 3 (on average 54 days after RT).

DARS were delineated on static MRI scans by one physician (CM), trained by a senior physician (SP). The MRI scans were acquired in axial, sagittal, and coronal slices in T2 (stir) sequence. T1 sequence ± gadolinium was done in axial and coronal (fat saturated) sections.

Contouring of DARS as recommended [13] was modified: the proximal part of the esophagus was defined in the sagittal slice in diameter and the base of tongue included the floor of mouth. The cervical part of the esophagus was excluded. As relevant organs for function of voice and prevention of aspiration, the supraglottic and transglottic larynx were defined as DARS (Fig 1).

All patients were treated by IMRT or helical TomoTherapy (n = 14 with VMAT (Rapid Arc™, Varian Medical Systems, Inc. Palo Alto, CA, USA; n = 3 with HI-ART TomoTherapy, Accuray, Madison, WI, USA).

The treatment parameters are given in Table 2.

All planning CT scans and RT plans were rigidly co-registered. For RT planning, a minimum of two planning CT scans is required–one prior to RT and a second at 40 Gy for boost planning. DARS were delineated on MRI and all planning CT scans. The treatment planning system calculates a plan sum dose volume histogram (DVH), which was read out for each single DARS. The mean dose ($D_{mean}$) representing the dose in 50% of the volume for each DARS was read out.

Table 1. Patient characteristics of all patients.

| Parameter | Patient characteristics |
|---|---|
| **Age** | mean 53.9 years [19–67] |
| **Sex** | female: n = 6 (35%); male: n = 11 (65%) |
| **Risk factors** | nicotine abuse (prior / ongoing) n = 12 (71%) |
| | alcohol addiction n = 10 (59%) |
| | poor hygiene and dental status n = 1 (6%) |
| **Histopathological findings** | squamous cell carcinoma n = 15 (88%) |
| | others n = 2 (12%) |
| **Tumor site** | oropharynx and oral cavity n = 8 (47%) |
| | hypopharynx n = 3 (17,6%) |
| | naso-oro-hypopharynx n = 3 (17,6%) |
| | oro-hypopharynx n = 2 (11,8%) |
| | parotid gland cancer n = 1 (5,9%) |
| **Tumor site of patients with FEES examination** | base of tongue n = 3 |
| | oro-hypopharynx n = 1 |
| | tonsillar fossa n = 1 |
| **TNM classification** (according to UICC 2010) | **T1: n = 1 (6%)** |
| | pT1b pN2c cM1(PUL) G1 |
| | **T2: n = 5 (29%)** |
| | pT2 pN1 cM0 G3 |
| | pT2 pN2a cM0 |
| | cT2 cN2b cM0 G3 |
| | cT2 cN3 cM0 G2 |
| | cT2 pN3 cM0 G3 |
| | **T3: n = 6 (35%)** |
| | pT3 pN2a cM0 G3 |
| | cT3 cN2b cM0 G2 |
| | cT3 cN2c cM0 G2 |
| | cT3 cN2c cM0 G3 |
| | cT3 cN2c cM0 G2 |
| | cT3 cN2c cM0 G3 |
| | **T4: n = 5 (29%)** |
| | cT4a cN2c cM0 G2 |
| | cT4a cN2c cM0 G2 |
| | cT4b pN2b cM0 G4 |
| | cT4 cN2c cM0 G2 |
| | cT4 cN3 cM0 G3 |
| **UICC-stage** (according to UICC 2010) | II: n = 1 |
| | III: n = 0 |
| | IVA: n = 8 |
| | IVB: n = 7 |
| | IVC: n = 1 |

In the second and prospective part of the study, five of the seventeen patients participated in a voice and swallowing test at the ENT department. The period between RT and examination was on average 22.2 months. For patient characteristics of the prospective cohort, see Table 3.

Three German language, standardized, and validated questionnaires were completed without assistance by the patients themselves for evaluation of subjective voice and swallowing

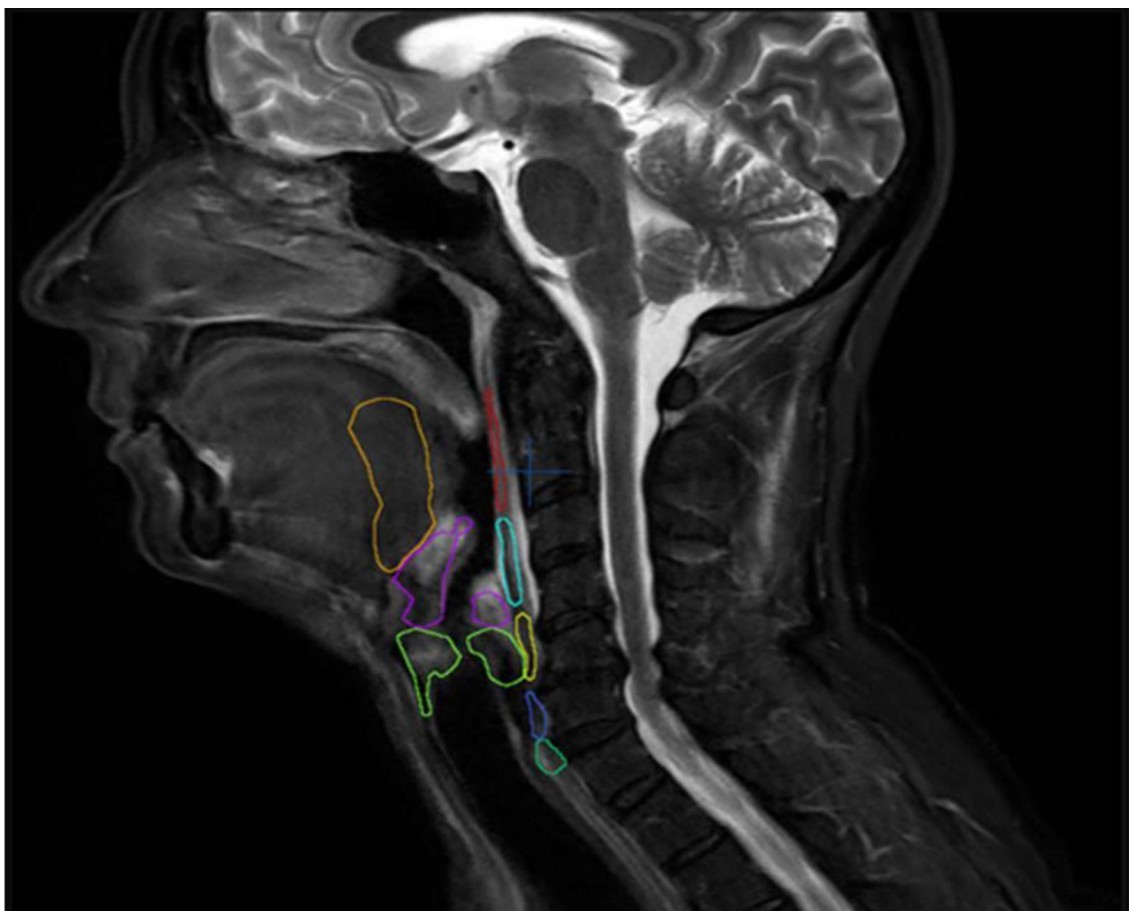

**Fig 1. DARS delineation on static MRI according to Christianen et al.** Overview of the eight OARs relevant for swallowing in sagittal MRI-slice (superior pharyngeal constrictor muscle (red), middle pharyngeal constrictor muscle (light blue), inferior pharyngeal constrictor muscle (yellow), crico-pharyngeal muscle (dark blue), proximal part of esophagus (dark green), posterior one third of the tongue and floor of mouth–base of tongue (orange), supraglottic larynx (purple) und transglottic larynx (light green).

disorders and as well as related quality of life (QoL), i.e. *Anderson Dysphagia Inventory (ADI-D)* to report dysphagia, the *Voice Handicap Index* (VHI) to report on dysphonia, and the questionnaire of the European Organisation for Research and Treatment of Cancer concerning quality of life in head and neck cancer patients (*EORTC QLQ- H&N 35)*. All questionnaires have been validated for the German language and are used routinely at the Ear, Neck and Throat (ENT) department of the Technical University of Munich.

Additionally, both a fiber-optic endoscopic evaluation of swallowing (FEES) with different consistencies and a voice test were performed. The procedure was performed by a senior consultant (SG) of the ENT department (TUM), who specialized in phoniatrics and pediatric audiology. *Rosenbeks 8-points-penetration-aspiration-scale (PAS)* [14, 15] was used to determine the severity of penetration and aspiration, while the Functional Oral Intake Scale (FOIS) classified by Crary [16] was employed to assess the oral food intake. The FOIS classification was done by the senior consultant (SG) of ENT department after FEES investigation. FEES is the standard procedure to detect disorders during the swallowing process by endonasal observation at the ENT department of TUM. Video fluoroscopy is performed only in some cases in order to avoid unnecessary x-ray stress to the patient. The voice test was carried out by a speech and language therapist. Objective assessment of the patient's phonation was done by

**Table 2. Information concerning treatment of all patients in the retrospective cohort.**

| Surgical Treatment | |
|---|---|
| Tumor resection | n = 6 (35%) |
| Neck dissection | n = 5 (29%) |
| **Radiooncological treatment** | |
| **Intent of radiotherapy** | |
| Definitive radiotherapy [total dose 70–70.4 Gy] | n = 12 |
| Adjuvant radiotherapy [total dose 64–64.2 Gy] | n = 4 |
| Definitive radiotherapy after neck dissection | n = 1 |
| **Combined modality treatment** | |
| Concomitant chemotherapy | n = 14 (82%) |
| **Technique of radiotherapy** | |
| Simultaneous integrated boost (by VMAT or TomoTherapy) | n = 5 |
| Volumetric Modulated Arc Therapy (VMAT) | n = 14 |
| TomoTherapy | n = 3 |
| **RT parameters** | |
| Median single dose | 2.0Gy [1.7–2.2Gy] |
| Median total dose | 70.0Gy [64.0–70.4Gy] |
| Median PTV (50Gy) | 1199.7cm$^3$ [591.6–1805.5cm$^3$] |
| Median duration of RT | 49 days [44–53 days] |
| | One patient was re-irradiated because of local recurrence (70.4 Gy as SIB in first course and 50.4Gy in second course, both with concomitant platin-based chemotherapy). |

automatic voice processing. Using this computer-aided analysis, the vocal range profile was determined. For classification of the severity of vocal disorder, the Dysphonia Severity Index (DSI) according to Wyuts [17] was applied. For graduation of classification systems mentioned previously, please see supplementary information.

## Ethics

Both the prospective study and final retrospective protocol were approved by the ethics committee of the medical faculty of the TUM (03/16/2016: 328/16 S). All procedures performed in this study involving human participants were in accordance with the ethical standards of the institutional and/or national research committee and with the 1964 Helsinki Declaration and its later amendments or comparable ethical standards. As for the retrospective analysis, all data was anonymized and the consent was waived by the ethics committee of the medical faculty of the TUM. For clinical investigation and questionnaires of the prospective cohort, the study protocol was amended by the ethics committee of the medical faculty of the TUM (07/21/2016: 328/16 S). Informed written consent was obtained from all five individual participants included in the prospective part of the study.

## Statistics

The retrospective analysis of DARS changes in MRI and CT over time were done on imaging datasets of 17 patients. For each patient, eight DARS were recorded at three different points of time on MRI, and dosimetric parameters were analyzed after co-registration with the planning

**Table 3. Patient characteristics of the five patients of the prospective cohort (SCC–squamous cell carcinoma, ECE- extracapsular extension of lymph nodes, VMAT–volumetric modulated arc therapy, MV–megavolt, SQB–sequential boost, SIB–simultaneous integrated boost, qd7 –repetition every seven days).**

| Prospective patient no. | No. 1 (female) | No. 2 (male) | No. 3 (female) | No. 4 (male) | No. 5 (male) |
|---|---|---|---|---|---|
| Age at diagnosis (yrs.) | 53 | 59 | 54 | 56 | 63 |
| Follow-up interval end of RT—examination | 4.26 yrs. | 3.6 yrs. and since Re-RT: 0.5 yrs. | 2.68 yrs. | 1.33 yrs. | 1.48 yrs. |
| Tumor localization | oropharynx (base of tongue) | Initial: oropharynx (base of tongue, tonsil, dorsal pharyngeal wall) <br><br> recurrence: dorsal pharyngeal wall | oropharynx (tonsil) | oropharynx (base of tongue) | oropharynx (base of tongue, tongue lateral side) |
| TNM stage | cT3 cN2c cM0 G3; SCC | Initial cT4a cN2c cM0 G2, SCC <br><br> Rec.: rcT2 rcN2c rcM0 G2, SCC | cT3 cN2c cM0 <br><br> SCC, p16 positive | pT3 pN2a (1/25ECE-) cM0 R1 G3, SCC | cT4a cN2c cM0 G2, SCC |
| Lung disease after RT | chron. bronchitis | no | no | No | Pneumonia |
| RT concept | definitive VMAT <br><br> 6 MV-photons, SQB <br><br> 2 Gy to 50 Gy <br><br> Boost 1: to 60 Gy <br><br> Boost 2: to 70 Gy | initial: definitive VMAT, 6 MV-photons, SIB <br><br> 1.7/2.0/2.2 Gy to 54.4/64.0/70.4Gy <br><br> recurrence: re-irradiation by TomoTherapy; <br><br> 6-MV-photons 1.8 Gy to 50.4 Gy (only FDG-PET-pos. tumor region and lymph nodes) | definitive VMAT <br><br> 6 MV-photons, SQB <br><br> 2 Gy to 50 Gy <br><br> Boost: to 70 Gy | adjuvant VMAT <br><br> 6 MV-photons, SQB 2 Gy to 50 Gy <br><br> Boost: to 66 Gy | definitive VMAT <br><br> 6 MV-photons, SQB 2 Gy to 50 Gy <br><br> Boost: to 70 Gy |
| Concomitant chemotherapy | Cisplatin 20 mg/m$^2$ <br><br> d1-5 and d29-33 | Initial: Cisplatin 20 mg/m$^2$ <br><br> d1-5 and d29-33 <br><br> Rec.: Cisplatin 40 mg/m$^2$ qd7; 5 cycles | Cisplatin 20 mg/m$^2$ d1-5 and d29-33 | Cisplatin 40 mg/m$^2$/d qd7 <br><br> 6 cycles | Cisplatin 40 mg/m$^2$/d qd7 <br><br> 5 cycles |
| Local tumor control | yes | no (3 yrs. after initial RT) <br><br> yes, after re-irradiation | yes | Yes | Yes |
| Neck Dissection | no | no | no | Yes | No |
| Feeding tube at time of swallowing investigation | no | no | no | No | Yes |

CT scan and RT dose matrix. Volume data of each DARS was calculated by iPlannet-software of Brainlab, Munich, Germany.

RT dose data was read out from the RT planning system Varian Medical Systems, Inc. Palo Alto, CA, USA 2013.

SPSS software (IBM SPSS Statistics version 21) was used for statistical analyses.

We calculated the volume difference between MRI 1 and MRI 2; MRI 2 and MRI 3 and MRI 1 and MRI 3. The paired t-test to analyze changes in the volume (differences) of the DARS and scatter plots were used for the explorative data analysis. The dose-volume relationship of the retrospective cohort was analyzed by linear regression. Also, a dose-dependence of dysphagia (measure–total applied dose) or dysphonia (measure total dose to the transglottic larynx) was analyzed. All information was extracted from patient charts. The analysis of this hypothesized relation was done by binary logistic regression. The level of significance p was set to $\leq 0.05$.

Only five of seventeen patients were able to take part in the prospective part of the study. Due to the small sample size of only five prospective investigated patients, the results of the swallowing and voice examinations are only statistically described.

## Results

### Retrospective part: Change of DARS volume

Static MRI 1 was defined as basic data set. The paired t-test showed a significant change in the volume of four DARS: superior pharyngeal constrictor (MRI 1 to 2: p = 0.035; MRI 1 to 3:

**Table 4. Binary logistic regression for influence of total dose (OAR) on development of dysphagia within the first 6 months and 6 to 12 months after the end of RT.**

|  | Regression coefficient β | Standard error | Wald | df | Sig. | Exp(β) |
|---|---|---|---|---|---|---|
| 1–6 months | 2.015 | 0.753 | 7.164 | 1 | 0.007 | 7.500 |
| 6–12 months | 1.540 | 0.636 | 5.863 | 1 | 0.015 | 4.667 |

p = 0.002), middle pharyngeal constrictor (MRI 1 to MRI 3: p = 0.019), cricopharyngeal muscle (MRI 1 of MRI 2: p = 0.38) and the proximal esophagus (MRI 1 of MRI 2: p = 0.16). There was a significant volume growth in three DARS (superior pharyngeal constrictor, middle pharyngeal constrictor, cricopharyngeal muscle) and a significant volume reduction in the proximal esophagus.

For each DARS, the mean dose ($D_{mean}$) was read out. The linear regression analysis could not detect any influence of $D_{mean}$ of DARS on volume change of DARS. Thus, there was no dose-dependent change of DARS volume in the 17 retrospective investigated patients.

Supraglottic and transglottic larynx as voice OARs (DARS) maintained a stable volume over time. Only the transglottic larynx showed a trend for change (p = 0.055).

Based on scatter plot analysis, there was no dose-related change in DARS volume over time.

## Dose-symptom relationship (swallowing and dysphonia)

All patient charts were checked for details on swallowing function and dysphonia (time points: pre-therapeutic, during RT and at follow-up).

The binary logistic regression (Table 3) showed a statistic significant (p = 0.007) 7.5 times higher risk for suffering from dysphagia within the first six months after RT when $D_{mean}$ to DARS is exceeded by steps of 1 Gy for all DARS. The risk of dysphagia six to 12 months after RT increases significantly (p = 0.015) by factor 4.7 with the same increment of exceeding mean dose ($D_{mean}$).

There was no dose dependency of $D_{mean}$ to the transglottic larynx and dysphonia (within the first six months: regression coefficient = 0.118; p = 0.808; Exp(ß) = 1.125 and from months six to twelve regression coefficient = -0.118; p = 0.808; Exp(ß) = 0.889, see Table 4). Furthermore, no correlation between volume changes, dysphagia, and dysphonia was found. The retrospective analysis was limited to notes in patient charts concerning normal or swallowing dysfunction and dysphonia and normal voice. Therefore, it was impossible to test whether DARS volume changes influenced the development of late swallowing and voice toxicities.

## Prospective part: Functional examination of voice and swallowing

Five of the thirteen surviving patients participated in prospective voice and swallowing tests. FEES described post radiogenic (edematous) morphological findings in three of the five patients and a reduced sensitivity in four of five patients (Table 5).

**Table 5. Results of FEES investigation.**

| Results of FEES | Patients n = 5 |
|---|---|
| Edema | 3 (60%) |
| reduced sensitivity | 4 (80%) |
| aspiration (liquid) | 2 (40%) |
| penetration (liquid / pulp) | 3 (60%) |
| Residuals | 3 (60%) |
| disorder of swallowing function grade 3 to 6 (FOIS scale) | 5 (100%) |

The functional swallowing test with pulpy consistency was classified as conspicuous in all five subjects (PAS 1–3). The functional test with liquids resulted in noticeable findings in three subjects with a PAS 2, 4 and 6. For solids (rusk), a considerable difficulty was seen in all subjects (PAS 4 and 6 in two subjects), which were compensated effectively. Residuals remained in the swallowing tract in three subjects. All subjects needed to drink after oral intake. In summary, disorder of swallowing function has been diagnosed in all subjects (grade 3 to 6 according to FOIS scale). Aspiration of liquids occurred in two subjects. Penetration was observed in three subjects (one with liquid and two with pulp). There was no penetration or aspiration of solids.

The evaluation of the patient reported Anderson Dysphagia Inventory–D (German) (ADI-D) showed an average total value of 59.2, which correlates to "rather conspicuous" in the Bauer and Rosanowski Scale.

The individual classification on the Penetration Aspiration Scale (PAS) evaluated by a senior physician of phoniatrics resulted in a total value of 48 ("very conspicuous"). Three subjects (total value of 55 to 61) had a "rather conspicuous" and another subject (total value of 76) reached a "rather inconspicuous" on the PAS system.

Dysphonia ranged from a mild to high degree at the Dysphonia Severity Index (DSI) scale.

The Voice Handicap Index (VHI) revealed a moderate voice disorder in one patient. Four had no subjective voice disorder. In the symptom scales of the EORTC QLQ-H & N35, only low scale scores were achieved for most categories (Fig 2).

Besides problems caused by xerostomia (dry (100%) and sticky saliva (63%)), only minor discomfort in respect to treatment related symptoms was reported by the majority of patients. All results of the prospective swallowing investigations are depicted in Table 6.

## Discussion

For detailed recontouring of DARS, static MRI is much more suitable than cine MRI [18]. Static MRI scans acquired at three different points of time were co-registered to the first (prior

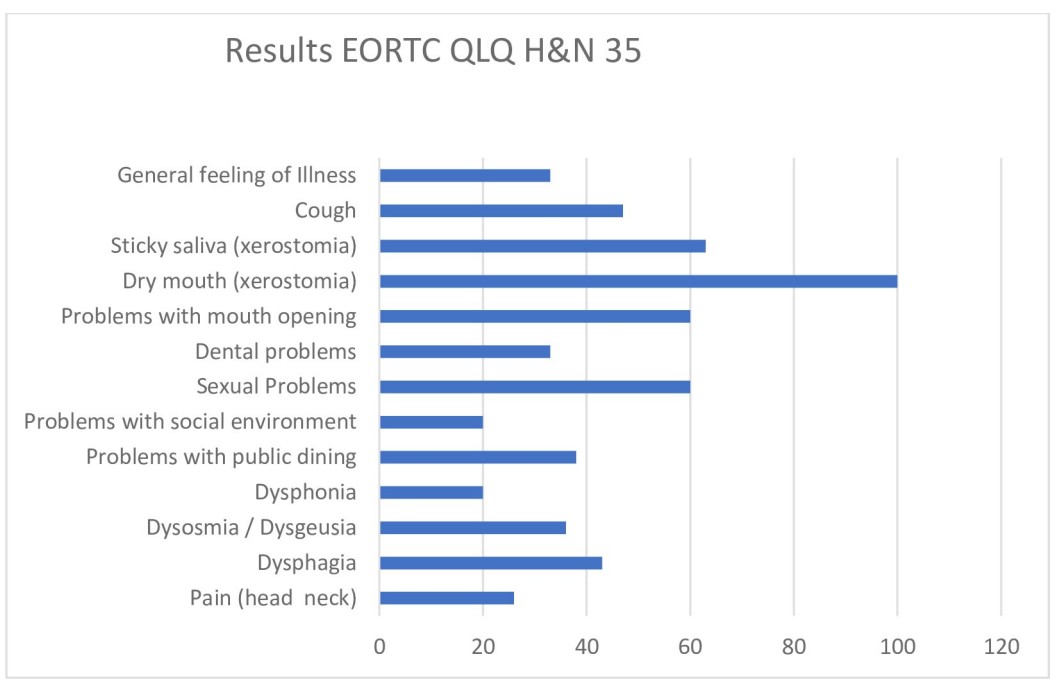

**Fig 2. Results of EORTC QLQ H&N35 questionnaire (results given in % of patients reporting these symptoms).**

**Table 6. Summary of results of the prospective investigations.**

| Parameter | Proband 1 | Proband 2 | Proband 3 | Proband 4 | Proband 5 |
|---|---|---|---|---|---|
| **Swallowing investigation** | | | | | |
| Morphology | post RT / edema | inconspicuous | post RT / edema | inconspicuous | post RT /edema |
| Sensibility | reduced | reduced | reduced | reduced | normal |
| FOIS Level | 4 | 6 | 5 | 5 | 3 |
| Aspiration | yes | yes | no | no | no |
| Penetration | no | yes | no | yes | yes |
| **Voice investigation** | | | | | |
| DSI | moderate dysphonia | moderate dysphonia | light dysphonia | slight dysphonia | profound dysphonia |
| **Questionnaires** | | | | | |
| ADI-D | rather conspicuous | rather inconspicuous | rather conspicuous | rather conspicuous | sure conspicuous |
| VHI | moderate dysphonia | no dysphonia | no dysphonia | no dysphonia | no dysphonia |
| EORTC QLQ-H&N35 Mean (%) | 72 | 38 | 34 | 38 | 41 |
| **Further Parameter** | | | | | |
| Logopedic training before investigation | yes | no | yes | no | No |
| Recommendation for logopedic training after investigation | yes | yes | no | yes | yes |
| Pulmonary symptoms or disease after RT | chron. bronchitis | no | no | no | pneumonia |
| Age at investigation | 57 | 63 | 57 | 57 | 65 |
| TNM at Diagnosis | T3 | T4+rcT2 | T3 | T3 | T4 |
| Interval RT—investigation [months] | 48 | 6 | 28 | 12 | 16 |
| Total dose RT [Gy] | 70 | 70,4 (+50,4 – reirradiation) | 70 | 66 | 70 |
| Local recurrence | no | yes | no | no | no |
| Neck dissection | no | no | no | yes | no |
| Impairment of movement | yes | no | yes | yes | yes |
| Feeding tube dependence at investigation | no | no | no | no | yes |

to start of RT) and the second (for boost RT) planning CT scans. T1 and T2 sequences were used for delineation of eight DARS as recommended [13].

Despite changes in volume in four of eight DARS over time (3/8 DARS increase and 1/8 decrease in volume), we did not find a correlation between DARS $D_{mean}$ and volume progressive change. In three DARS (superior pharyngeal constrictor, middle pharyngeal constrictor, cricopharyngeal muscle), we noticed a gradual increase in volume. Ricchetti et al. showed by recontouring OARS on weekly repeated planning CT scans over seven weeks of RT in 91.6% of their 26 patients with oropharyngeal cancer an enlargement of the volume of the constrictor muscles [19] and other OARs until week five. They mentioned that once a significant change in volume over the baseline was recognized, this remained for the following weeks. In our investigation, the last MRI was performed six weeks post RT, supporting the results of Ricchetti et al. During the time we treated the patients of our retrospective analysis, we did not use special dose constraints for DARS. Popovtzer et al. showed a $D_{mean} > 60$ Gy to the constrictor muscles will change the volume of these muscles detected by MRI three months post RT [20]. This group pointed towards pathophysiological changes in soft tissue of OARs in twelve different head and neck cancer patients, indicating that underlying effects of inflammation and consecutive edema were in fact a result of acute irradiation induced damage. Also, in our patients, both inflammation and edema could provide an explanation for the increase in DARS volume over time.

Fast swallowing function registration can be done by cine MR or video fluoroscopy as objective investigation methods. Video fluoroscopy, despite X-ray exposure of the patient, is the gold standard of objective assessment of dysphagia. In contrast, FEES is an endonasal procedure performed by a physician without exposing the patient to radiation.

The retrospective analysis was based on notes in patient charts concerning dysphagia and dysphonia. $D_{mean}$ to DARS revealed an increased risk of dysphagia by a factor of 7.5 for the first six months post RT for every additional Gray above average $D_{mean}$ to DARS. For the interval six to twelve months post RT, this factor drops to 4.6 for every additional applied Gray of average $D_{mean}$ to DARS. Thus, the risk of developing dysphagia immediately after the end of RT up to six months thereafter is higher than in the second half year post RT. Perhaps, this is a finding caused by prolonged acute dysphagia as a consequence of radiation induced pharyngeal mucositis and consecutive edema. In 2007, Levendag et al. presented data concerning $D_{mean}$ of RT to superior and middle constrictor muscle inducing severe dysphagia. They found a steep dose-effect relationship for late dysphagia: "increased probability of dysphagia of 19% with every additional 10 Gy" [21]. Patients of this trial were treated by 3D-conformal radiotherapy or IMRT and in some cases, with a brachytherapy boost. Dysphagia was assessed by patient reported outcome with EORTC H&N-35 and ADI questionnaires. For DARS delineation, the authors used the planning CT scans.

$D_{mean}$ of DARS had no influence on the development of dysphonia in our trial. Despite whole-field IMRT for oropharyngeal or cancer of unknown primary, Sanguineti et al. reported only mild voice changes determined by RT dose to the larynx. The calculated threshold was Larynx $D_{mean} \leq 50$ Gy [22].

Charters et al. performed a systematic review and meta-analysis of the impact of dosimetry to DARS [23]. They elucidated that despite the many papers published on this topic, the results on dose-effect relationships are neither homogeneous nor are the underlying investigation methods heterogeneous. It is a matter of fact that beam path dose of RT will touch DARS and lead to acute and late side effects. Especially for long term survivors, this will pose a struggle for years to come. For instance, Peponi et al. described the long-term results and treatment outcome for a group of patients with cancer of the larynx, oropharynx, and hypopharynx. They implemented a midline protection contour for RT planning from below the hyoid to the 2/3 cervical vertebra in order to reduce dose to the constrictor muscle. This midline contour was drawn outside the PTV. The 5-year local control rate was 75% and no treatment failure at the region of the midline protection contour was detected. Thirty-two months after RT, grade 3 and 4 toxicities were 10% [24]. Reducing side effects by sparing OARs in general while ensuring that target volumes receive the prescribed dose is the advantage of IMRT and should be the standard of care in head and neck cancer RT. Particularly parotid gland sparing leads to remarkable progress in reducing the incidence of xerostomia and aids swallowing function by better gliding of food [10].

In terms of local tumor control for patients with locally advanced squamous cell head and neck cancer, the ESCALOX trial [25] hypothesized a benefit of 15% at 2 years when a dose escalation up to 80.5 Gy to the GTV of the primary tumor and lymph nodes ($\geq$ 2cm) is applied, while taking into account protection of OAR [26]. The literature review of Charters et al. [23] displays the inhomogeneity of DARS, the different dose parameters to DARS, and heterogeneous use of important functional outcome parameters (i.e. aspiration and stricture of DARS) analyzed in detail of 23 studies. Despite these obvious limitations, the authors conclude that impairment of the constrictor pharyngeal muscles is important for development of dysphagia. A radiotherapy dose threshold of 50 Gy is given for constrictor pharyngeal muscles. Despite protection of DARS, it is important to prescribe and apply a tumoricidal RT dose to the cancer. Sometimes, this may be detrimental to the swallowing function, but it is a crucial

factor in saving patients´ lives. From the mentioned 23 studies [23], only 11 used gold standard examination methods (video fluoroscopy or FEES) to classify swallowing disorders.

For our objective evaluation, five patients participated in a voice and swallowing test.

In general, patients assessed the voice and swallowing function subjectively better than the observer. Other authors observed similar results, discussing an accommodation of the patient to the dysfunction of swallowing [27]. In contrast to subjective patient rating, FEES examination revealed in 3 of 5 patients a disorder of penetration and aspiration for liquids. Disorder of swallowing was not recognized by the patient. For pulpy and solid consistency, an impairment was detected in all patients. A dysphagia FOIS level 3–6 stresses these results. Two patients suffered from aspiration and three other patients were diagnosed with penetration (PAS). In a previous meta-analysis of deglutition disorders extracted from 17 studies including 229 patients, Porto de Toledo et al. reported a high frequency of aspiration in 28.6% of patients immediately posttreatment and during the following three months [28]. For the course of six months posttreatment, this meta-analysis revealed a penetration of fluids above the vocal folds and an impairment of laryngeal elevation. In contrast to our patients, in all studies, a baseline pretreatment imaging examination for swallowing (video fluoroscopy or fiber optic endoscopic evaluation of swallowing) was available. All kinds of RT techniques, fractionation schedules and chemotherapy schemes were included. The pretreatment investigations confirmed an aspiration in 8.4% of all meta-analysis patients. In only five studies, the pretreatment status concerning penetration of food, liquids or saliva was determined with a frequency of 10.5%. The penetration rate increased posttreatment up to 33.6% at the time point longer than six months posttreatment. This fact is limited since only three studies had observed this parameter. Aspiration during the first three months posttreatment was 28.6% and declined in the following three months to 17.6%. Later, the aspiration frequency was 16.2%. As for safety, the increasing percentage of patients with pharyngeal residues from less than six months (47.1%) to six months posttreatment (61.8%) is alarming. A higher frequency of patients with pharyngeal residues was reported after a follow-up more than six months with 73.8%, but only four out of 17 studies presented data at this point of time. Pharyngeal residues should be taken seriously, because they are precursors of aspiration. In summary, a higher frequency of deglutition disorders posttreatment compared to baseline parameters was demonstrated in cases of multimodality treatment.

From the studies analyzed within the meta-analysis [28], only the trial of Patterson et al. [29] used FEES as we did. They observed in 8 of 97 patients a silent aspiration at three months posttreatment. Only two groups presented data using the penetration aspiration scale (PAS). At our head and neck cancer center, FEES with PAS classification is the standard of care in routine examination today.

In all patients, a posttreatment change of speech and voice (mild to severe) was diagnosed.

Heijnen et al. reviewed literature until 2016 concerning dysphagia, speech, and voice changes after radioncological treatment of HNC patients. They reported an improvement of voice after RT, but disorders of swallowing and oral intake. Quality of life was also reduced after RT [30]. Finally, the authors mentioned the heterogeneous group of patients and the diversity of applied examination methods to describe functional outcome after RT. It is hard to compare results of different trials because of heterogeneous measures.

In the prospective part of our trial, five patients participated in the Anderson Dysphagia Inventory (ADI-D) and the Voice Handicap Index (VHI) for subjective patient assessment in German language. For ADI-D, an average result of 59.2 points was calculated, indicating a rather conspicuous swallowing function posttreatment. In contrast to the objective voice analysis, only in one case did a patient report moderate dysphonia. For patient reported quality of life after head and neck cancer treatment (EORTC QLQ-H & N35), only low scores (1 –no

impairment, 2 –little impairment; 3 –moderate impairment; 4 –strong impairment) were achieved for most categories. As known from routine follow-up examinations, all patients complained about xerostomia (63% about sticky saliva, 43% mentioned swallowing impairment).

The RT of head and neck cancer is associated with side effects reducing QoL for patients during and post therapy. Late RT-induced toxicity caused by morphological and functional changes of swallowing structures impairs QoL. Many patients are not aware of the protective function of coughing during swallowing. Because of loss of cough reflex due to reduced sensitivity after radiation, caused for instance by fibrosis or ulceration, patients suffer from silent aspiration and falsely consider themselves safe because of misinterpretation of cough. Without this reflex, penetration and aspiration is not recognized. It is important to be cognizant of this pathology, because aspiration can induce pneumonia. In some cases, patients die due to aspiration-induced pneumonia, especially long-term survivors (mortality 20–65%) [31]. Therefore, Eisbruch and colleagues developed the concept of DARS (dysphagia / aspiration related structures) [32] by using CT images early in the IMRT era. Knowledge about pathophysiology and development of deglutition disorders is important in order to prevent patients developing detrimental sequelae. FEES and video-fluoroscopy offer real-time insights into the swallowing process.

The retrospective investigations of DARS of 17 HNC patients contoured on static MRI and co-registered to CT-based RT plans were complemented by prospective clinical swallowing and voice examinations. Unfortunately, only five patients could be investigated prospectively from this group. This small sample size rendered it impossible to safely estimate dose-function effects.

## Conclusion

By co-registration of static MR to planning CT scans, it was possible to delineate DARS better than with CT imaging alone. Over the course of RT and posttreatment of HNC, an increase in DARS volume was detected. With only 17 patients in the retrospective part, it was impossible to clarify the dosimetric impact on swallowing function posttreatment. Five prospectively investigated patients had changes in swallowing function, despite the fact that most patients did not recognize swallowing dysfunction.

## Supporting information

**S1 Data.**
(XLSX)

**S1 File. List of abbreviations.**
(DOCX)

**S1 Table. Grading of ADI-D.**
(DOCX)

**S2 Table. 8-point-penetrations-aspirations-scale (PAS) by Rosenbek [13].**
(DOCX)

**S3 Table. Scale of oral food intake (FOIS) by Crary et al. [15].**
(DOCX)

**S4 Table. Classification of dysphonia by using DSI-values defined by Wyuts [23].**
(DOCX)

## Acknowledgments

The authors wish to thank all patients who participated in the study and the team at the pho-niatric department of the Ear, Nose and Throat Department at the Technical University of Munich. We thank Dr. Sabrina Renfro-Kohl for her kind revision of the English manuscript and Dr. Victoria Kehl for her kind check of the statistical data.

## Author Contributions

**Conceptualization:** Steffi U. Pigorsch, Kerstin A. Kessel.

**Data curation:** Chaline May, Simone Graf.

**Formal analysis:** Chaline May, Birgit Waschulzik.

**Investigation:** Steffi U. Pigorsch, Chaline May, Simone Graf.

**Resources:** Henning Bier.

**Supervision:** Kerstin A. Kessel, Simone Graf, Fridtjof Nüsslin, Stephanie E. Combs.

**Writing – original draft:** Steffi U. Pigorsch.

**Writing – review & editing:** Steffi U. Pigorsch.

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
