## [Decision Letter · Decision Letter 0]

30 Dec 2019

PONE-D-19-32133

MRI-determined Changes of Dysphagia / Aspiration related Structures (DARS) during and after Radiotherapy: influence on Function and Quality of Life (QoL) of Head and Neck Cancer Patients

PLOS ONE

Dear Dr. Kessel, 

Thank you for submitting your manuscript to PLOS ONE. After careful consideration, we feel that it has merit but does not fully meet PLOS ONE’s publication criteria as it currently stands. Therefore, we invite you to submit a revised version of the manuscript that addresses the points raised during the review process.

ACADEMIC EDITOR: The study is interesting. Please kindly respond to the questions raised by the reviewers.

We would appreciate receiving your revised manuscript by Feb 13 2020 11:59PM. To enhance the reproducibility of your results, we recommend that if applicable you deposit your laboratory protocols in protocols.io, where a protocol can be assigned its own identifier (DOI) such that it can be cited independently in the future. For instructions see: http://journals.plos.org/plosone/s/submission-guidelines#loc-laboratory-protocols

We look forward to receiving your revised manuscript.

Kind regards,

Jason Chia-Hsun Hsieh, M.D. Ph.D

Academic Editor

PLOS ONE

Additional Editor Comments (if provided):

The study is interesting. Please kindly respond to the questions raised by the reviewers.

Journal Requirements:

2. Please note that PLOS journals require authors to make all data underlying the findings described in their manuscript fully available without restriction at the time of publication. When specific legal or ethical requirements prohibit public sharing of a dataset, authors must indicate how researchers may obtain access to the data. PLOS journals will not consider manuscripts for which the following factors influence ability to share data:

- Authors will not share data because of personal interests, such as patents or potential future publications.

- The conclusions depend solely on the analysis of proprietary data, whether these data are owned by the authors, by their funders or institutions, or by other parties.

Therefore, please update your Data Availability statement to indicate how other researchers may gain access to the underlying data reported in the manuscript. For more information, please see: https://journals.plos.org/plosone/s/data-availability.

3. Please provide additional details regarding participant consent. In the ethics statement in the manuscript, you have specified that 'informed consent was obtained from all individual participants included in the study'. Please clarify whether informed consent was obtained from 17 patients to use their medical records in the retrospective study, or whether consent was obtained from the 5 patients to participate in the prospective study. If informed consent was obtained for both parts of the study, please specify this in both the ethics statement in the manuscript and the online submission form.

If consent was only obtained for 5/17 patients for the prospective part of the study, please clarify whether the patient records of the remaining 12/17 patients were analyzed anonymously, or whether consent was waived by the ethics committee.

Additionally, in the ethics statement in the Methods and online submission information, please ensure that you have specified what type of consent you obtained (for instance, written or verbal, and if verbal, how it was documented and witnessed). If your study included minors, state whether you obtained consent from parents or guardians.

4. To comply with PLOS ONE submission guidelines, in your Methods section, please provide additional information regarding your statistical analyses. For more information on PLOS ONE's expectations for statistical reporting, please see https://journals.plos.org/plosone/s/submission-guidelines.#loc-statistical-reporting.

5. At this time, we ask that you please clarify whether the 'Ethics Committee of the Medical Faculty TUM' specifically reviewed and approved both the retrospective and prospective parts of the present study.

6. In your Methods section, please provide additional information about the participant recruitment method and the demographic details of your participants used in the prospective study. Please ensure you have provided sufficient details to replicate the analyses such as: a) the recruitment date range (month and year), b) a description of any inclusion/exclusion criteria that were applied to participant recruitment, c) a description of how participants were recruited, and d) descriptions of where participants were recruited and where the research took place.

7. Please include additional information regarding the questionnaires (ADI-D, Voice-H-I, EORTC QLQ-H&N35 Mean) used in the study and ensure that you have provided sufficient details that others could replicate the analyses. For instance, if you developed a questionnaire as part of this study and it is not under a copyright more restrictive than CC-BY, please include a copy, in both the original language and English, as Supporting Information.

Reviewers' comments:

Reviewer's Responses to Questions

**Comments to the Author**

1. Is the manuscript technically sound, and do the data support the conclusions?

Reviewer #1: No

Reviewer #2: Yes

Reviewer #3: No

2. Has the statistical analysis been performed appropriately and rigorously? 

Reviewer #1: No

Reviewer #2: I Don't Know

Reviewer #3: No

3. Have the authors made all data underlying the findings in their manuscript fully available?

Reviewer #1: Yes

Reviewer #2: Yes

Reviewer #3: No

4. Is the manuscript presented in an intelligible fashion and written in standard English?

Reviewer #1: Yes

Reviewer #2: Yes

Reviewer #3: No

5. Review Comments to the Author

Reviewer #1: The authors performed a retrospective study of 17 patients with head and neck cancers to investigate the usefulness of MRI before, during and after radiation therapy.

Although the focus of this paper seems to be important in future radiotherapy, this paper has some serious problems.

First of all, there were only 17 cases in this study. In addition, this study included tumors at various sites and stages. There were also various treatments for tumors. The possibility that these tumor sites and treatments affected dysphagia, dysphonia, or other symptoms cannot be ruled out, and it is difficult to say that the scientific validity of the results of this study is guaranteed.

The authors should increase the number of cases and unify the tumor site and treatment.

Other remarks are provided below.

1) The introduction should be a little shorter. For some sentences, the authors should move them to the discussion part.

2) I think that abbreviations should be unified. Please consider whether to write abbreviations in the text or at the end of the manuscript.

3) The authors should unify the description of date. “1.1.2008” or “5/2012”

Reviewer #2: This is a well written paper regarding MRI-based evaluation of Dysphagia/Aspiration Related Stractures(DARS).

Among 780 patients with head and neck cancer, 17 patients who had MRI before RT (MRI 1), at 40Gy (MRI 2) and 6 weeks after RT (MRI 3) were selected and retrospectively analyzed after the DARS were delineated on MRI and CT.

Furthermore, as a prospective part of the study, the authors carried out voice and swallowing tests in five of the 17 patients.

As the results, they found statistically significant changes in volume of four DARS; including increases and a decrease in volume depending on the sub-component of the DARS. However, there was no dose dependence of total dose to DARS in the alternation of volumes.

From the symptom described on the patient charts, dysphagia was related to the mean dose of DARS, but there was no significant relationship between volume changes of DARS and dysphagia.

Impression:

The authors investigated well in anatomical and functional changes related to DARS. However, they seemed to fail in constructing a definite conclusion from the findings they observed, although each finding is valuable itself. It will be acceptable for publication if the authors describes more of other related studies regarding this theme, and make it clear that what’s new this study(studies) could be added to the current shared knowledges.

Questions:

How the author selected 5 patients who participate in prospective study from the 17 patients?

.

Minor points

Page 4, paragraph 2, last line: A full stop (and perhaps some preceding words) is missing.

Reviewer #3: The goal of the authors of the manuscript is to the potential of MR-image guidance for RT of head and

neck tumors because of better discrimination and resolution of soft tissues as DARS in MRI compared to CT.

The manuscript is poorly written and the authors do not respect the basic rules of scientific writing. Some sentences are impossible to understand due to the apparent absence of knowledge of some rules of english writing

The authors are not consistent in their use of acronyms. acronyms must be defined first completely in english with acronyms written between parenthesess after the definition

In a scientific paper the authors must always assume that their readers do not know fully the content of the paper and thus must be crystal clear. They must "take the readers gently by the hand" to present their scientific goal, to explain the materials and methods used, to justify the statistical methods used, to display clearly their data and then discuss seriously theirs results and especially the limitations of their study if there are any.

1) This abstract is unacceptable full of acronyms not previously defined

The table of ACRONYMS is at the end of the manuscript !!!!!!!!!!

Integration of MRI into RT-planning gives the opportunity to delineate DARS more precisely

for RT. DARS are important for QoL in H&N patients since late effects cause dysfunction. By

MRgRT improved DARS-contouring and adaption during RT is possible. 17 H&N-patients

(treated 5/2012 - 8/2015) were analyzed retrospectively. 14/17 had concomitant chemotherapy.

Median time RT- phoniatric evaluation was 1.78 years [range 0.42-4.26]. Patients were treated

by IMRT median single dose: 2Gy [1.7-2.2Gy] and cumulative dose: 70Gy [64-70.4Gy]. DARS

were delineated on MRI and planning CT scans and co-registered with all RT plans. 5/17

patients participated in a prospective voice and swallowing test (last RT – examination: 22.2

months (average)). For PRO Anderson Dysphagia Inventory, Voice Handicap Index and

EORTC QLQ- H&N 35 were applied.

FEES, voice test and automatic voice processing were used for objective assessment.

There was neither a dose dependence of Dmean DARS volume-changes over time nor of

dysphonia and no correlation between volume changes, dysphagia or dysphonia. One additional

Gy on Dmean DARS causes a 7.5fold risk to suffer from early (first 6 months after end of RT)

dysphagia and 4.7fold later than 6 months. By FEES 3/5 patients were diagnosed with post

radiogenic changes of morphology and 4/5 with reduced sensitivity. Functional swallowing test

detected disturbances in all cases. Dysphonia had a high variation. Swallowing related QoL

was "rather conspicuous". Patients ranked themselves good in EORTC QLQ-H & N35.

Every additional Gy on Dmean of DARS increases the risk for late dysphagia and should be

avoided.

2) It is not acceptable that the authors do not make available extensively their data about patients for so-called ethical reasons. The anonymisation of patients is always possible and garantee the private life and the identity of all patients

3) It is not clear how the 17 patients includec in this study were chosen out of more 700 patients

4) The introduction is very poor

5) the Materials and Methods are not clear at all. There is none justification for any of the statistical method used. A Specialist in Biostatistics should be included among the authors

6) The inclusion of various tumors at different stages with very few patients for each stage is a big flaw of this study

7) The classification of the tumors is not explicit for any common reader

8) The presentation of the results is very confusing

9) the discussion must be completely be rewritten

A dramatic revision is mandatory

6. PLOS authors have the option to publish the peer review history of their article (what does this mean?). If published, this will include your full peer review and any attached files.

Reviewer #1: Yes: Eiichiro Okazaki

Reviewer #2: No

Reviewer #3: No

---

## [Author Response · Author response to Decision Letter 0]

6 Apr 2020

ACADEMIC EDITOR: The study is interesting. Please kindly respond to the questions raised by the reviewers.

We would appreciate receiving your revised manuscript by Feb 13 2020 11:59PM. NOTE: Please note, we thank You very much for extension of deadline until March 20th 2020.

 Additional Editor Comments (if provided):

The study is interesting. Please kindly respond to the questions raised by the reviewers.

Journal Requirements:

 Answer: We followed these recommendations.

2. Please note that PLOS journals require authors to make all data underlying the findings described in their manuscript fully available without restriction at the time of publication. When specific legal or ethical requirements prohibit public sharing of a dataset, authors must indicate how researchers may obtain access to the data. PLOS journals will not consider manuscripts for which the following factors influence ability to share data:

- Authors will not share data because of personal interests, such as patents or potential future publications.

- The conclusions depend solely on the analysis of proprietary data, whether these data are owned by the authors, by their funders or institutions, or by other parties.

Therefore, please update your Data Availability statement to indicate how other researchers may gain access to the underlying data reported in the manuscript. For more information, please see: https://journals.plos.org/plosone/s/data-availability.

Answer: We will make data available in anonymized form as separate file. 

3. Please provide additional details regarding participant consent. In the ethics statement in the manuscript, you have specified that 'informed consent was obtained from all individual participants included in the study'. Please clarify whether informed consent was obtained from 17 patients to use their medical records in the retrospective study, or whether consent was obtained from the 5 patients to participate in the prospective study. If informed consent was obtained for both parts of the study, please specify this in both the ethics statement in the manuscript and the online submission form.

If consent was only obtained for 5/17 patients for the prospective part of the study, please clarify whether the patient records of the remaining 12/17 patients were analyzed anonymously, or whether consent was waived by the ethics committee.

Additionally, in the ethics statement in the Methods and online submission information, please ensure that you have specified what type of consent you obtained (for instance, written or verbal, and if verbal, how it was documented and witnessed). If your study included minors, state whether you obtained consent from parents or guardians.

Answer: We have explained in detail the ethics statement and the patient information as well as the procedure for patient informed consent in the Methods section.

 4. To comply with PLOS ONE submission guidelines, in your Methods section, please provide additional information regarding your statistical analyses. For more information on PLOS ONE's expectations for statistical reporting, please see https://journals.plos.org/plosone/s/submission-guidelines.#loc-statistical-reporting.

Answer: We have fully complied with the submission guidelines and affirm that statistical analyses have been performed by the responsible statistician and cross-checked by yet another statistician. 

5. At this time, we ask that you please clarify whether the 'Ethics Committee of the Medical Faculty TUM' specifically reviewed and approved both the retrospective and prospective parts of the present study.

Answer: Firstly, the Ethics Committee of the Medical Faculty TUM reviewed and approved the retrospective part of the study. The prospective part of the study was reviewed and proved as well by the Ethics Committee of the Medical Faculty TUM with an amended version of the initial trial protocol. 

6. In your Methods section, please provide additional information about the participant recruitment method and the demographic details of your participants used in the prospective study. Please ensure you have provided sufficient details to replicate the analyses such as: a) the recruitment date range (month and year), b) a description of any inclusion/exclusion criteria that were applied to participant recruitment, c) a description of how participants were recruited, and d) descriptions of where participants were recruited and where the research took place.

Answer: We have thoroughly covered the questions raised in the Methods section.

 7. Please include additional information regarding the questionnaires (ADI-D, Voice-H-I, EORTC QLQ-H&N35 Mean) used in the study and ensure that you have provided sufficient details that others could replicate the analyses. For instance, if you developed a questionnaire as part of this study and it is not under a copyright more restrictive than CC-BY, please include a copy, in both the original language and English, as Supporting Information.

Answer: All questionnaires are standardized and will be uploaded as a supplement. At no time was a questionnaire of our own making employed in the study. 

Reviewers' comments:

Reviewer's Responses to Questions

NOTE: Please note, our answers are included in yellow.

1. Is the manuscript technically sound, and do the data support the conclusions?

Reviewer #1: No

Reviewer #2: Yes

Reviewer #3: No

Answer: We have subjected the manuscript to a complete and thorough revision and verification as recommended by the third reviewer.________________________________________

2. Has the statistical analysis been performed appropriately and rigorously? 

Reviewer #1: No

Reviewer #2: I Don't Know

Reviewer #3: No

 Answer: The complete statistical analysis has been revised anew by two statisticians.________________________________________

3. Have the authors made all data underlying the findings in their manuscript fully available?

Reviewer #1: Yes

Reviewer #2: Yes

Reviewer #3: No

Answer: We will, of course, make our anonymized raw data available.________________________________________

 4. Is the manuscript presented in an intelligible fashion and written in standard English?

Reviewer #1: Yes

Reviewer #2: Yes

Reviewer #3: No

Answer: The manuscript has been corrected by a native speaker with a language training background.

5. Review Comments to the Author

Reviewer #1: The authors performed a retrospective study of 17 patients with head and neck cancers to investigate the usefulness of MRI before, during and after radiation therapy.

Although the focus of this paper seems to be important in future radiotherapy, this paper has some serious problems.

First of all, there were only 17 cases in this study. In addition, this study included tumors at various sites and stages. There were also various treatments for tumors. The possibility that these tumor sites and treatments affected dysphagia, dysphonia, or other symptoms cannot be ruled out, and it is difficult to say that the scientific validity of the results of this study is guaranteed.

The authors should increase the number of cases and unify the tumor site and treatment.

Answer: As the discussion indicates, there are only very few studies on this topic comprising a larger number of patients. All in all, this patient collective largely proves to be non-compliant. Not all patients are willing to tolerate the considerable amount of expenditure involved in conducting the necessary examinations. The small number of 17 retrospective patients is due to the required entirety of DARS imaging. Only patients with complete DARS imaging were included. All patients still alive at the time of the prospective study were invited. However, only five of these patients agreed to participate in the evaluation. We plan to carry out further swallowing examinations in a prospective follow-up study.

Other remarks are provided below.

1) The introduction should be a little shorter. For some sentences, the authors should move them to the discussion part.

Answer: We have revised and shortened the Introduction section.

2) I think that abbreviations should be unified. Please consider whether to write abbreviations in the text or at the end of the manuscript.

• All abbreviations have been named and set in parentheses after the first use of the term.

3) The authors should unify the description of date. “1.1.2008” or “5/2012”

Answer: All date descriptions have been uniformed.

Reviewer #2: This is a well written paper regarding MRI-based evaluation of Dysphagia/Aspiration Related Stractures(DARS).

Among 780 patients with head and neck cancer, 17 patients who had MRI before RT (MRI 1), at 40Gy (MRI 2) and 6 weeks after RT (MRI 3) were selected and retrospectively analyzed after the DARS were delineated on MRI and CT.

Furthermore, as a prospective part of the study, the authors carried out voice and swallowing tests in five of the 17 patients.

As the results, they found statistically significant changes in volume of four DARS; including increases and a decrease in volume depending on the sub-component of the DARS. However, there was no dose dependence of total dose to DARS in the alternation of volumes.

From the symptom described on the patient charts, dysphagia was related to the mean dose of DARS, but there was no significant relationship between volume changes of DARS and dysphagia.

Impression:

The authors investigated well in anatomical and functional changes related to DARS. However, they seemed to fail in constructing a definite conclusion from the findings they observed, although each finding is valuable itself. It will be acceptable for publication if the authors describes more of other related studies regarding this theme, and make it clear that what’s new this study(studies) could be added to the current shared knowledges.

Answer: In the last two years, other studies on this subject were compiled in various reviews due to the fact that the individual studies comprised only a small number of patients. These investigations were examined in detail in both the review and the individual publications. 

Questions:

How the author selected 5 patients who participate in prospective study from the 17 patients?

Answer: All surviving patients at the time of our prospective study were invited. However, only five agreed to participate in the evaluation.

Minor points

Page 4, paragraph 2, last line: A full stop (and perhaps some preceding words) is missing.

Answer: The manuscript has been fully revised as recommended by Reviewer 3. 

Reviewer #3: The goal of the authors of the manuscript is to the potential of MR-image guidance for RT of head and neck tumors because of better discrimination and resolution of soft tissues as DARS in MRI compared to CT.

The manuscript is poorly written and the authors do not respect the basic rules of scientific writing. Some sentences are impossible to understand due to the apparent absence of knowledge of some rules of english writing

Answer: We have taken this criticism to heart and implemented necessary changes and improvements in both our general and scientific style of writing. We have been careful to observe English language standards pertaining to correct spelling, punctuation, and common usage. 

The authors are not consistent in their use of acronyms. acronyms must be defined first completely in english with acronyms written between parenthesess after the definition

Answer: We have ensured a thorough revision concerning this point as well.

In a scientific paper the authors must always assume that their readers do not know fully the content of the paper and thus must be crystal clear. They must "take the readers gently by the hand" to present their scientific goal, to explain the materials and methods used, to justify the statistical methods used, to display clearly their data and then discuss seriously theirs results and especially the limitations of their study if there are any.

Answer: We can affirm that we are well aware of this goal and have adapted our manuscript accordingly so that its contents can be fully understood and appreciated by non-professionals as well. 

1) This abstract is unacceptable full of acronyms not previously defined

Answer: We would like to politely call to your attention that a table of acronyms can be found as a supplementary file. 

The entire content has not only been adapted to all PlosOne requirements, but completely revised as well.

Integration of MRI into RT-planning gives the opportunity to delineate DARS more precisely

for RT. DARS are important for QoL in H&N patients since late effects cause dysfunction. By

MRgRT, (comma) improved DARS-contouring and adaption during RT is possible. 17 H&N-patients

(treated 5/2012 - 8/2015) were analyzed retrospectively. 14/17 had concomitant chemotherapy.

Median time RT- phoniatric evaluation was 1.78 years [range 0.42-4.26]. Patients were treated

by IMRT median single dose: 2Gy [1.7-2.2Gy] and cumulative dose: 70Gy [64-70.4Gy]. DARS

were delineated on MRI and planning CT scans and co-registered with all RT plans. 5/17

patients participated in a prospective voice and swallowing test (last RT – examination: 22.2

months (average)). For PRO Anderson Dysphagia Inventory, Voice Handicap Index and

EORTC QLQ- H&N 35 were applied.

FEES, voice test and automatic voice processing were used for objective assessment.

There was neither a dose dependence of Dmean DARS volume-changes over time nor of

dysphonia and no correlation between volume changes, dysphagia or dysphonia. One additional

Gy on Dmean DARS causes a 7.5fold risk to suffer from early (first 6 months after end of RT)

dysphagia and 4.7fold later than 6 months. By FEES 3/5 patients were diagnosed with post

radiogenic changes of morphology and 4/5 with reduced sensitivity. Functional swallowing test

detected disturbances in all cases. Dysphonia had a high variation. Swallowing related QoL

was "rather conspicuous". Patients ranked themselves good in EORTC QLQ-H & N35.

Every additional Gy on Dmean of DARS increases the risk for late dysphagia and should be

avoided.

2) It is not acceptable that the authors do not make available extensively their data about patients for so-called ethical reasons. The anonymisation of patients is always possible and garantee the private life and the identity of all patients

Answer: All data will be made accessible as anonymized raw data.

3) It is not clear how the 17 patients includec in this study were chosen out of more 700 patients

Answer: The precise method pertaining to the choice of patients from a total number of 780 patients was clearly explained and can be found in the Methods section. 

4) The introduction is very poor

Answer: The Introduction has been revised in its entirety.

5) the Materials and Methods are not clear at all. There is none justification for any of the statistical method used. A Specialist in Biostatistics should be included among the authors

Answer: A statistician is amongst the authors. Moreover, the data has since been examined twice by other statisticians. Early on, the presented analysis had already been evaluated by our statistician.

6) The inclusion of various tumors at different stages with very few patients for each stage is a big flaw of this study

Answer: This is certainly the case. Patient selection for this study is limited, because complete MRI imaging of DARS is a prerequisite and vital criterion for inclusion in the study. MRIs were not performed for the purpose of DARS imaging, but for reasons of portrayal of the tumor and lymph nodes with the aim of contouring the target volume for radiation therapy. This explains the heterogeneity of the examined tumor patients.

7) The classification of the tumors is not explicit for any common reader

Answer: For details and explanation of tumor classification, we recommend further reading at ww.UICC.org.

8) The presentation of the results is very confusing

Answer: The presentation of the results has been exhaustively restructured.

9) the discussion must be completely be rewritten

Answer: Both the discussion and the entire paper have been rewritten. We thank you kindly for your valuable comments.

A dramatic revision is mandatory

Answer: This has indeed been implemented. 

6. PLOS authors have the option to publish the peer review history of their article (what does this mean?). If published, this will include your full peer review and any attached files.

Answer: We want to publish the whole history of review.

Do you want your identity to be public for this peer review? For information about this choice, including consent withdrawal, please see our Privacy Policy.

Reviewer #1: Yes: Eiichiro Okazaki

Reviewer #2: No

Reviewer #3: No

---

## [Decision Letter · Decision Letter 1]

11 May 2020

PONE-D-19-32133R1

MRI-determined Changes of Dysphagia / Aspiration-Related Structures (DARS) during and after Radiotherapy: Influence on Function and Quality of Life (QoL) of Head and Neck Cancer Patients

PLOS ONE

Dear Pigorsch,

Thank you for submitting your manuscript to PLOS ONE. After careful consideration, we have decided that your manuscript does not meet our criteria for publication and must therefore be rejected.

Specifically:

ACADEMIC EDITOR: Although the topic is interesting, two of the three reviewers gave a "rejection" to the manuscript. The two reasons include: (a) the too-small sample size and poor statistical power; (b) This small sample size rendered it impossible to estimate dose-function effects safely. A case series/report might be suitable for publication. 

I am sorry that we cannot be more positive on this occasion, but hope that you appreciate the reasons for this decision.

Yours sincerely,

Jason CH Hsieh, M.D. Ph.D

Academic Editor

PLOS ONE

Additional Editor Comments (if provided):

Although the topic is interesting, two of the three reviewers gave a "rejection" to the manuscript. The two reasons include: (a) the too-small sample size and poor statistical power; (b) This small sample size rendered it impossible to estimate dose-function effects safely. A case series or reports might be suitable for publication.

Reviewers' comments:

Reviewer's Responses to Questions

**Comments to the Author**

1. If the authors have adequately addressed your comments raised in a previous round of review and you feel that this manuscript is now acceptable for publication, you may indicate that here to bypass the “Comments to the Author” section, enter your conflict of interest statement in the “Confidential to Editor” section, and submit your "Accept" recommendation.

Reviewer #1: All comments have been addressed

Reviewer #2: All comments have been addressed

Reviewer #3: (No Response)

2. Is the manuscript technically sound, and do the data support the conclusions?

Reviewer #1: No

Reviewer #2: Yes

Reviewer #3: No

3. Has the statistical analysis been performed appropriately and rigorously? 

Reviewer #1: No

Reviewer #2: Yes

Reviewer #3: No

4. Have the authors made all data underlying the findings in their manuscript fully available?

Reviewer #1: Yes

Reviewer #2: Yes

Reviewer #3: Yes

5. Is the manuscript presented in an intelligible fashion and written in standard English?

Reviewer #1: Yes

Reviewer #2: Yes

Reviewer #3: Yes

6. Review Comments to the Author

Reviewer #1: I confirmed the revised paper and the authors' response to the reviewer's comments.

I pointed out that the number of subjects in this study was very small (17 patients), which is not enough to obtain significant results. I think the authors' response to this indication is reasonable. Accumulation of a sufficient number of cases requires a lot of labor and cost. However, this fact does not make up for the small number of cases, and I think that the fact that sufficient cases have not been obtained in statistical studies is a huge disadvantage. I believe that the authors' responses are not sufficient answers regarding this statistical disadvantage. If there is not sufficient statistical proof, I recommend increasing the number of cases or writing a paper in the form of a case report.

Reviewer #2: (No Response)

Reviewer #3: The authors have made significant uimprovements to their manuscript and provided transparently their data.

Nevertheless, The number of quite different patients at different stages of their tumors enrolled in this study is a major flaw of this investigation.

The title and goal of this manuscript is "MRI-determined Changes of Dysphagia / Aspiration-Related Structures (DARS) during and after Radiotherapy: Influence on Function and Quality of Life (QoL) of Head and Neck Cancer Patients". This study was supposed to support the hypothesis according to which Function and Quality of Life (QoL) of Head and Neck Cancer Patients are significantly improved. This goal could not be reached with such a very low number of patients who participated to this study.

The co-registration of static MR to planning CT scans, made possible a better delineation of DARS than with CT imaging alone. Over the course of RT and posttreatment of HNC, an increase in DARS volume was detected. This result could be very easily expected. The retrospective investigations of DARS of 17 HNC patients contoured on static MRI and co-registered to CT-based RT plans were complemented by prospective clinical swallowing and voice examinations. Unfortunately, only five patients could be investigated prospectively from this group. This small sample size rendered it impossible to safely estimate dose-function effects. This study does add any grounbreaking information to the field of clinical and basic oncology.

7. PLOS authors have the option to publish the peer review history of their article (what does this mean?). If published, this will include your full peer review and any attached files.

Reviewer #1: Yes: Eiichiro Okazaki

Reviewer #2: No

Reviewer #3: No

- - - - -

---

## [Author Response · Author response to Decision Letter 1]

9 Jun 2020

Rebuttal Letter on PONE-D-19-32133R1

Dear Reviewer, dear colleagues,

Please find attached our answers to the questions and concerns of the reviewers to our re-submitted manuscript.

Thank You for Your support.

Yours sincerely,

Prof. Dr. med. S.E. Combs Dr.med. St. U. Pigorsch

Head of Department Managing Senior Physician

Comments to the Author

1. If the authors have adequately addressed your comments raised in a previous round of review and you feel that this manuscript is now acceptable for publication, you may indicate that here to bypass the “Comments to the Author” section, enter your conflict of interest statement in the “Confidential to Editor” section, and submit your "Accept" recommendation.

Reviewer #1: All comments have been addressed

Reviewer #2: All comments have been addressed

Reviewer #3: (No Response)

Answer: We would like to center Your attention to above reviewer comments.

2. Is the manuscript technically sound, and do the data support the conclusions?

Reviewer #1: No

Reviewer #2: Yes

Reviewer #3: No

Answer: For further explanation please see our answer to each reviewer comment. 

3. Has the statistical analysis been performed appropriately and rigorously? 

Reviewer #1: No

Reviewer #2: Yes

Reviewer #3: No

Answer: We refer to our response to the reviewer comments upon submission of the revised manuscript: “The complete statistical analysis has been revised anew by two statisticians.”

We must emphasize the status of our second statistician, who was independent and not involved in the initial analysis. This data review and statistics check was done especially in light of the reviewer comments. 

4. Have the authors made all data underlying the findings in their manuscript fully available?

Reviewer #1: Yes

Reviewer #2: Yes

Reviewer #3: Yes

5. Is the manuscript presented in an intelligible fashion and written in standard English?

Reviewer #1: Yes

Reviewer #2: Yes

Reviewer #3: Yes________________________________________

6. Review Comments to the Author

Reviewer #1: I confirmed the revised paper and the authors' response to the reviewer's comments.

I pointed out that the number of subjects in this study was very small (17 patients), which is not enough to obtain significant results. I think the authors' response to this indication is reasonable. Accumulation of a sufficient number of cases requires a lot of labor and cost. However, this fact does not make up for the small number of cases, and I think that the fact that sufficient cases have not been obtained in statistical studies is a huge disadvantage. I believe that the authors' responses are not sufficient answers regarding this statistical disadvantage. If there is not sufficient statistical proof, I recommend increasing the number of cases or writing a paper in the form of a case report.

Answer: Neither labor nor the cost of an extended study were the reasons for the limited patient number. We explained in detail how the number of 17 patients was reached. MR scans representing all DARS completely at each point of time were our primary focus. Unfortunately, we were able to only extract 17 patients from a total of 780 with this inclusion criteria. The follow-up data on DARS organ function of these 17 patients were gathered from patient charts retrospectively.

Retrospective analysis in general are impaired by availability of complete datasets or images. We have referred to this limitation and the constricted conclusions. The findings of the five prospective examined patients for their swallowing function after therapy were only statistically described. 

However, the main part of the manuscript deals with the 17 retrospective patients investigated for their DARS organ changes over a longer course of time including radiotherapy treatment weeks and follow-up time.

Reviewer #2: (No Response)

Reviewer #3: The authors have made significant improvements to their manuscript and provided transparently their data.

Nevertheless, The number of quite different patients at different stages of their tumors enrolled in this study is a major flaw of this investigation.

Answer: The presented data was generated from a retrospective analysis focusing on morphologic and volumetric changes of swallowing structures over the course of radiotherapy and later during follow-up by MR and planning CT scans. One goal was to improve the delineation of DARS by co-registration of MR scans to the planning CT scans for better anatomic discrimination. The other aim was to demonstrate and prove the hypothesis that DARS were changed over time by means of radiotherapy in regards to volume and structure. The last point – the influence of radiotherapy dose to the amount of DARS volume and function changes - could not be proven safely because only 17 of the investigated patients fulfilled the inclusion criteria. 

If a prospective study for DARS evaluation were designed, the inclusion criteria would be: tumor entity, tumor stage, treatment intention (definitive or postoperative radiotherapy), combination with systemic therapy, imaging: CT and MR scans at defined time points, base line, and follow-up investigation of swallowing and voice function.

The title and goal of this manuscript is "MRI-determined Changes of Dysphagia / Aspiration-Related Structures (DARS) during and after Radiotherapy: Influence on Function and Quality of Life (QoL) of Head and Neck Cancer Patients". This study was supposed to support the hypothesis according to which Function and Quality of Life (QoL) of Head and Neck Cancer Patients are significantly improved. This goal could not be reached with such a very low number of patients who participated to this study.

Answer: Improvement of QoL for HNC patients was not the goal of this study. 

The aims of our study were clearly defined at the conclusion of the INTRODUCTION: “The aim of this work was to assess the changes of DARS during and after radiooncological treatment of HNC by co-registration of static MRI and planning CT scans for better delineation of DARS in order to detect morphological changes over the course of RT. The second aim was to compare these dosimetric and volumetric data to results of functional investigation of swallowing and voice.”

The title of the submitted manuscript is obviously misleading for the reviewer. We like to apologize for this. To focus on the main part of the investigation we changed the title.

From: "MRI-determined Changes of Dysphagia / Aspiration-Related Structures (DARS) during and after Radiotherapy: Influence on Function and Quality of Life (QoL) of Head and Neck Cancer Patients".

To: "MRI- and CT-determined Changes of Dysphagia / Aspiration-Related Structures (DARS) during and after Radiotherapy of Head and Neck Cancer".

For our explanation concerning the number of patients, please see above.

The co-registration of static MR to planning CT scans, made possible a better delineation of DARS than with CT imaging alone. Over the course of RT and posttreatment of HNC, an increase in DARS volume was detected. This result could be very easily expected. The retrospective investigations of DARS of 17 HNC patients contoured on static MRI and co-registered to CT-based RT plans were complemented by prospective clinical swallowing and voice examinations. Unfortunately, only five patients could be investigated prospectively from this group. This small sample size rendered it impossible to safely estimate dose-function effects. This study does add any groundbreaking information to the field of clinical and basic oncology.

Answer: For our explanation concerning the number of patients, please see above. 

7. PLOS authors have the option to publish the peer review history of their article (what does this mean?). If published, this will include your full peer review and any attached files.

---

## [Editor Report · Decision Letter 2]

29 Jul 2020

MRI- and CT-determined Changes of Dysphagia / Aspiration-Related Structures (DARS) during and after Radiotherapy

PONE-D-19-32133R2

Dear Dr. Pigorsch,

We’re pleased to inform you that your manuscript has been judged scientifically suitable for publication and will be formally accepted for publication once it meets all outstanding technical requirements.

Kind regards,

Jessica D McDermott, MD, MSCS and Qinghui Zhang

Academic Editors

PLOS ONE

---

## [Editor Report · Acceptance letter]

14 Aug 2020

PONE-D-19-32133R2 

MRI- and CT-determined Changes of Dysphagia / Aspiration-Related Structures (DARS) during and after Radiotherapy 

Dear Dr. Pigorsch:

I'm pleased to inform you that your manuscript has been deemed suitable for publication in PLOS ONE. Congratulations! Your manuscript is now with our production department. 

Kind regards, 

on behalf of

Dr. Jessica D McDermott 

Academic Editor

PLOS ONE